# Chemical Diversity and Biological Activity of Secondary Metabolites Isolated from Indonesian Marine Invertebrates

**DOI:** 10.3390/molecules26071898

**Published:** 2021-03-27

**Authors:** Fauzia Izzati, Mega Ferdina Warsito, Asep Bayu, Anggia Prasetyoputri, Akhirta Atikana, Linda Sukmarini, Siti Irma Rahmawati, Masteria Yunovilsa Putra

**Affiliations:** Research Center for Biotechnology, Indonesian Institute of Sciences, Jl. Raya Jakarta-Bogor KM 46 Cibinong, Bogor, West Java 16911, Indonesia or fauz012@lipi.go.id (F.I.); mferdina@gmail.com (M.F.W.); anggia.prasetyo@gmail.com (A.P.); akhirta@gmail.com (A.A.); linda.sukmarini@lipi.go.id (L.S.); rahmawatisitiirma@gmail.com (S.I.R.)

**Keywords:** Indonesia, marine natural product, marine invertebrates, soft corals, sponges, tunicates, biodiversity, biological activity

## Abstract

Marine invertebrates have been reported to be an excellent resource of many novel bioactive compounds. Studies reported that Indonesia has remarkable yet underexplored marine natural products, with a high chemical diversity and a broad spectrum of biological activities. This review discusses recent updates on the exploration of marine natural products from Indonesian marine invertebrates (i.e., sponges, tunicates, and soft corals) throughout 2007–2020. This paper summarizes the structural diversity and biological function of the bioactive compounds isolated from Indonesian marine invertebrates as antimicrobial, antifungal, anticancer, and antiviral, while also presenting the opportunity for further investigation of novel compounds derived from Indonesian marine invertebrates.

## 1. Introduction

A wide range of natural products (NPs) has been isolated from various marine organisms, especially marine invertebrates such as sponges, tunicates, soft corals, bryozoans, and nudibranchs. These marine invertebrates are excellent sources of NPs with vast chemical structures and potential biological activities [1,2,3]. During 2012–2017, no less than 550–700 new compounds have been reported from marine invertebrates [4], in which half of these compounds were isolated from marine sponges [5]. Among those, 4% and 22% of the compounds were identified in 2017 and 2016, respectively [4,5]. Between 1998 to 2018, one hundred and fourteen secondary metabolites were isolated from the marine sponges of the genus *Suberea* [6]. Meanwhile, a hundred and seventy compounds were isolated from soft corals of the genus *Dendronephthya* alone throughout 1999–2019 [7]. Soft corals belonging to the genus *Xenia* are rich in terpenoids, with 199 compounds isolated from 1977–2019 [8]. To date, approximately 30,000–40,000 marine natural products (MNPs) have been identified, with the majority of the compounds exhibiting cytotoxic and anticancer properties [4,5,9]. 

The biological potential of MNPs from marine invertebrates has been proven to be a valuable source for drug discovery and development. Most of the approved commercial marine-based drugs are of marine invertebrate origin. Eight marine drugs have been approved by the US Food and Drug Administration (FDA) and European Medicines Agency (EMA). The first approved marine drug is Ziconotide (Prialt^®^) discovered in the marine snail *Conus magus*. This peptide-derived marine drug is commonly used as an analgesic drug for the management of severe chronic pain through the intrathecal route. The second one is Omega-3-acid-ethyl esters (Lovaza^®^) derived from fish oil and used as an anti-hypertriglyceridemia drug. The third is Vidarabine (Vira-A^®^) derived from the sponge *Cryptotethya crypta*, while the fourth is Iota-carrageenan (Carragelose^®^), derived from red macroalgae. These two were registered as antivirals. The other four drugs were approved for cancer treatment, i.e., (1) *viz.* Cytarabine (Cytosar-U^®^ and Depocyt^®^) discovered in sponge *Cryptotethya crypta*, (2) Trabectedin (Yondelis^®^) isolated from the tunicate *Ecteinascidia turbinata*, (3) Eribulin mesylate (Halaven^®^) discovered from sponge *Halichondria okadai*, and (4) Brentuximab vedotin (Adcetris^®^) derived from the sea hare *Dolabella auricularia*. Additionally, approximately 30 MNPs were reported in different clinical trial stages, mainly derived from marine invertebrates [9,10,11].

Indonesia is the world’s largest archipelagic country with 17,500 islands and a long coastline of 81,000 km. This extraordinary geographic attribute offers a highly diverse variety of marine organisms, resulting in Indonesia being known as a mega-biodiversity of marine organisms. Like other living organisms, marine species synthesize metabolites, either primary or secondary, to support their lives. Living in an extreme environment often induces these organisms to synthesize multiple secondary metabolites with unique chemical properties. A part of their defense mechanism, these metabolites have also been reported to have diverse biological activities that are important for drug discovery and development [12]. 

The study of Indonesian MNPs was started in 1972 by Engelbrecht et al., who discovered a compound called 25-hydroxy-24ξ-methylcholesterol derived from a soft coral collected from Nias Island, Indonesia [13]. Following that, Cornery et al. reported two novel cytotoxic compounds from the Indonesian sponge *Hyatella* sp. called *viz.* laulimalide and isolaulimalide, which were active against the KB cell line with an IC_50_ value of 15 ng/mL [14]. Since then, research on Indonesian MNPs has expanded significantly. From the 1970s to the year 2017, about 732 MNPs were isolated from Indonesian sea waters. They were mainly produced by sponges (Porifera), tunicates (Chordata), and soft corals (Cnidaria) [15]. In the past decade, hundreds of novel compounds have been discovered from Indonesian marine organisms, many of which showing potent biological activity [16,17].

This review presents recent updates on Indonesian MNPs isolated from three marine invertebrates (sponges, tunicates, and soft corals), reported from 2007 to 2020, covering the chemical diversity and biological activity.

## 2. Marine Invertebrates

### 2.1. Sponges

Among sponges, alkaloids were reported as the most isolated bioactive compounds, followed by terpenoids, peptides, and polyketides (Table 1). Among the isolated alkaloids from sponges, manzamines are mostly reported to exhibit a broad spectrum of biological activities such as cytotoxic, antimicrobial, antimalarial, antiviral, anti-inflammatory, antiatherosclerotic, insecticidal, and proteasome inhibitor [18]. Their structure has a fused tetra- or pentacyclic ring attached to a carboline moiety.

To date, more than 80 menzamine-derived alkaloids have been isolated from sponges. This report showed 21 alkaloids were isolated from marine sponge *A. ingens,* five of which were identified as novel manzamine alkaloids, acanthomanzamines A-E (**1–5**). These five acanthomanzamines were isolated from the marine sponge *A. ingens* collected in Mantehage, North Sulawesi [18]. The acanthomanzamines A (**1**) and B (**2**) were the first compounds to contain the 1,2,3,4-tetrahydroisoquinoline-6,7-diol moiety instead of the β-carboline moiety, while the acanthomanzamine C (**3**) contains the hexahydrocyclopenta[*b*]-pyrrol-4(2*H*)-one ring system, and the acanthomanzamines D (**4**) and E (**5**) have the oxazolidine and two methyloxazolidine rings, respectively (Figure 1). In terms of biological activity, the acanthomanzamines **1** and **2** demonstrated more potent cytotoxic activity against cervical cancer HeLa cells (IC_50_ values 4.2 and 5.7 µM, respectively) than the acanthomanzamines **4** and **5** (IC_50_ values 15 and >20 µM, respectively). However, the acanthomanzamines **4** and **5** showed better proteasome inhibition against the proteasomal chymotrypsin-like activity (IC_50_ values of 0.63 and 1.5 µM, respectively). These findings suggested that the presence of 1,2,3,4-tetrahydroisoquinoline-6,7-diol probably enhanced the cytotoxicity. Meanwhile, compounds containing β-carboline perform better on chymotrypsin-like activity. Additionally, the acanthomanzamines **1, 2, 4,** and **5** inhibited the accumulation of the cholesterol ester at 20 µM in macrophages with 48%, 73%, 73%, and 61%, respectively [18]. 

The sponge *A. ingens* also produces other types of nitrogen-containing molecules, i.e., acantholactam (**6**), pre-*neo*-kauluamine (**7**), acathocyclamine A (**8**), *epi*-tetradehydrohalicyclamine B (**9**), tetradehydrohalicyclamine B (**10**), halicyclamine B (**11**), chloromethylhalicyclamine B (**12**), cyclo (d-Pro-l-Phe) (**13**), cyclo (l-Pro-Gly) (**14**), cyclo (l-Pro-l-Ala) (**15**), cyclo (d-Pro-l-Val) (**16**), cyclo (l-Pro-Ser) (**17**), cyclo (d-Pro-l-Ile) (**18**), and cyclo (l-Pro-l-Tyr) (**19**). Alkaloid compounds **6** and **7** were isolated from *A. ingens* collected in Bajotalawaan, North Sulawesi [19]. Compound **7** exhibited proteasome inhibitory activity with an IC_50_ value of 0.34 µM, whereas compound **6** showed little to no activity [19]. These findings indicated that the eight-membered ring in the manzamines plays a key role in their bioactivity. 

A novel 3-alkylpiperidine alkaloid (**8**) was also isolated from *A. ingens* collected from Wakatobi Marine National Park in Southeast Sulawesi [20]. Recently, the alkaloid compounds **9**–**19** were also isolated from *A. ingens* collected from South Sulawesi along with compound **8** [21]. Moreover, compound **8** was reported to have specific antimicrobial activity against *E. coli* and showed an inhibitory effect on amyloid β-42 production induced by aftin-5 without cytotoxicity at 26 µM. Compounds **8** and **11** exhibited antibacterial activity at 100 µg/disc against *E. coli* and *S. aureus*. In addition to that, compound **12** was reported to have selective inhibitory activity against the protein kinase CK1 δ/ε with an IC_50_ value of 6 µM, while the diketopiperazine compound **13** showed a selective kinase inhibitory activity against CDK2/cyclin A with an IC_50_ value of 1 µM. However, no bioactivity was reported from compounds **14**–**19 [21]**. 

A study reported two new pyrimidine-β-carboline alkaloids—namely, ingenines C (**20**) and D (**21**), successfully isolated from *A. ingens* obtained in Sulawesi [22]. The ingenines **20** and **21** exhibited cytotoxic activities towards the human breast MCF-7 and colorectal HCT116 cancer cells (IC values of 4.33 and 6.05 µM, and 2.90 and 3.35 µM, respectively) [22].

Two bromopyrrole alkaloids, dispacamide E (**22**) and ethyl 3,4-dibromo-1*H*-pyrrole-2-carboxylate (**23**), were isolated from methanolic extract of the marine sponge *S. massa* collected in Papua, Indonesia [23]. Compound **22** had significant protein kinase inhibitory activities against GSK-4, DYRKIA, and CK-1 with IC_50_ values of 2.1, 6.2, and 4.9 µM, respectively [23]. Another four new alkaloids were isolated from genus *Stylissa* in Derawan Island, Berau, Northeast Kalimantan—namely, 12-*N*-methylstevensine (**24**), 12-*N*-methyl-2-debromostevensine (**25**), 3-debromolatonduine B methyl ester (**26**), and 3-debromolatonduine A (**27**). Among these four compounds, however, only compound **24** showed significant *in-vitro* cytotoxic activity against the mouse lymphoma L5187Y cancer cell line with an EC_50_ value of 8.7 µM [24].

Recently, three new guanidine alkaloids, crambescidins 345 (**28**), 361 (**29**), and 373 (**30**), were isolated from *C. bulbotoxa* collected in Samalona Island, South Sulawesi [25]. Interestingly, crambescidin **28** was the first crambescidin analogue with a non-alkylated tetrahydropyrane ring (Figure 2). Crambescidin **29** possesses a rare propyl substituent and another tetrahydropyrane ring in place of the left-side unsaturated seven-membered ring. Meanwhile, crambescidin **30** was the first analogue with an ethyl group at the right-side tetrahydropyrane ring, which possesses a methyl group in most reported crambescidin-type alkaloids. These three guanidine alkaloids (**28, 29,** and **30**) exhibited moderate cytotoxicity against the human epidermoid A431 carcinoma cell line with IC_50_ values of 7.0, 2.5, and 0.94 µM, respectively [25]. They also showed anti-oomycete activity against the oomycete plant pathogen *Phytophthora capsici* [25].

Four new imidazole alkaloids were isolated from the Indonesian sponge *L. chagosensi* (Table 1). The alkaloids methyldorimidazole (**31**) and preclathridine B (**32**) were isolated from Kapoposang Island, South Sulawesi [26], whereas the other two alkaloids, naamidines H (**33**) and I (**34**), were isolated from *L. chagosensi* collected in North Sulawesi [27,28]. Compounds **33** and **34** were reported to show cytotoxicity against HeLa cells with IC_50_ values of 11.3 and 29.6 µM, respectively [27,28]. Another new alkaloid was isolated from *L. microraphis,* named spironaamidine (**35**) [28]. The spironaamidine possessed antimicrobial activity against *B. cereus*. However, the effect was slightly lower than that of naamidine **33** [27,28], probably due to the presence of spiroquinone, which is exceptionally rare in natural sources (Figure 3).

Two new β-carboline alkaloids, variabines A (**36**) and B (**37**), were isolated from Indonesian marine sponge *Luffariealla variabilis* collected in North Sulawesi [29]. Compound **36** was the first sulfated β-carboline alkaloid and the sulfonated derivative of compound **37**. A hydroxy group in compound **37** replaces the sulfate group at C-6 in compound **36**. Compound **37** was found to inhibit the chymotrypsin-like activity of the proteasome and Ubc13 (E2)-Uev1A interaction with IC_50_ values of 16.5 and 20.6 µM, respectively. In contrast, compound **36** showed a weak effect on proteasome. This result suggested that the sulfate group in variabines decreases chymotrypsin-like activity [29].

A novel pyridoacridine alkaloid, sagitol C (**38**), was isolated from the ethyl acetate fraction of *Oceaniapia* sp. collected in Ambon, Maluku Islands [30]. Alkaloid **38** was found to exhibit cytotoxic activity against the mouse lymphoma L5187Y cancer cell line, HeLa cell, and the rat pheochromocytoma PC12 cell lines with ED_50_ values of 0.7, 0.9, and 2.3 µM, respectively [30].

Seven novel anti-angiogenic steroidal alkaloids (Figure 4) were isolated from the sponge *Corticium simplex* collected in Flores Island, East Nusa Tenggara [31,32], namely cortistatins E-H (**39**–**42**) and J-L (**43**–**45**). These compounds have unique abeo-9(10-19)-stigmastane-type steroidal alkaloids. Compounds **38**–**42** have oxabicyclo[3.2.1]octene and *N-*methyl piperidine or 3-methylpyridine units in the side chain, while compounds **43**–**45** have an isoquinoline unit. Compound **43** showed cytostatic antiproliferative activity against human umbilical vein endothelial cells (HUVECs) at 8 nM [31,32].

The genus *Aaptos* is also an abundant source of novel aaptamine alkaloids (Table 1). Four new aaptamine derivates were isolated from *A. suberitoides* collected in Ambon, Maluku Islands [33]—namely, 11-methoxy-3*H*-[1,6]naphthyridino [6,5,4-*def*]quinoxalin-3-one (**46**), 2,11-dimethoxy-3*H*-[1,6]naphthyridino [6,5,4-*def*]quinoxalin-3-one (**47**), 5-benzoyldemethylaaptamine (**48**), and 3-aminodemethyl(oxy)aaptamine (**49**). Compound **47** is the 11-methoxy derivative of compound **46** and represents a benzoyl-aaptamine skeleton that has not been previously identified [33]. Concerning the biological activity, compound **48** exhibited cytotoxic activity against the mouse lymphoma L5187Y cancer cell line with an IC_50_ value of 5.5 µM [33]. Another aaptamine derivate isolated from *Aaptos* sp. collected from Kupang, East Nusa Tenggara [34],—namely, 2-methoxy-3-oxoaaptamine (**50**), demonstrated an anti-mycobacterial activity under both aerobic and hypoxic conditions, both with a MIC value of 23 µM [34].

Twenty-one novel psammaplysin derivatives were successfully isolated from *A. strongylata* collected in Tulamben Bay, Bali [35]. Compounds 19-Hydroxypsammaplysin E (**51**), psammaplysin K (**52**), psammaplysin K dimethoxy acetal (**53**), psammaplysin L (**54**), and M (**55**), possess modified aromatic ring substituents. Meanwhile, psammaplysin N-P (**56**–**58**), 19-hydroxypsammaplysin P (**59)**, psammaplysin Q (**60**), 19-hydroxypsammaplysin Q (**61**), psammaplysin R-S (**62–63**), 19-hydroxypsammaplysin S (**64**), psammaplysin T (**65**), and 19-hydroxypsammaplysin T (**66**) exhibit various side chains that are saturated fatty acid side chains (Figure 5). On the other hand, psammaplysin U (**67**), 19-hydroxypsammaplysin U (**68**), psammaplysin V-W (**69**–**70**), and 19-hydroxypsammaplysin W (**71**) have monoenoic fatty acid side chains. Among these 21 compounds, 19-hydroxypsammaplysin E (**51**), having the *N*-substitution of the ethylamino moiety, showed the best antimalarial activity with modest inhibition against *P. falciparum* (IC_50_ = 6.4 µM) [35].

Several new terpenes were isolated from Indonesian marine sponges from the Dictyoceratida order (Figure 6), such as *L. herbacea* [36], *Spongia* sp. [37], and *Carterospongia foliascens* [38]. Four new sesquiterpenes were isolated from *L. herbacea* collected in Manado, North Sulawesi—namely, lamellodysidines A (**72**) and B (**73**), *O*,*O*-dimethyllingshuiolide A (**74**), and 11-*epi*-*O*,*O*-dimethyllingshuiolide A (**75**) [36]. Compound **72** was the first compound identified with a unique bridged polycyclic framework, and compound **73** is a novel nitrogenous sesquiterpene, while compounds **74** and **75** were obtained as an inseparable mixture due to their 11-epimeric nature [36].

Sponges from the genus *Spongia* are known as a rich source of unique terpenoids (Table 1). This is evident in the successful isolation of three novel terpenoids from *Spongia* sp. collected in Bunaken Marine Park, North Sulawesi [37]. These three compounds were identified as diterpene—namely, 18-nor-3,17-dihydroxyspongia-3,13(16),14-trien-2-one (**76**), 18-nor-3,5,17-trihydroxyspongia-3,13(16),14-trien-2-one (**77**), and spongiapyridine (**78**). Compound **77** possesses an unusual d-ring substitute, a pyridyl ring system, in the place of δ-lactone (Figure 6). In terms of the biological activity of these four compounds, only compound **76** possessed moderate activity on aromatase inhibition with an IC_50_ of 34 µM and quinone reductase 1 induction with the concentration needed to double the enzymatic response of 11.2 µM [37].

Four new scalarane-based sesterterpenoids were isolated from *C. foliascens* associated with the coral reefs of Palau Barang Lompo, near Makasar, South Sulawesi [38]. The closely related compounds were identified as 20,24-bishomo-25-norscalaranes 1 (**79**) and 2 (**80**), and 20,24-bishomoscalaranes ketals 3 (**81**) and 4 (**82**) (Figure 6). Compounds **81** and **82** were isolated as an inseparable mixture. Compound **80** showed little to no activity toward RCE-protease inhibition compared to **79**. Meanwhile, a combination of **80** and **82** (plus **81,** as its inseparable mixture) had inhibition activity with IC_50_ values of 38 and 4.2 µg/mL, respectively [38]. Compound **79** and the mixture of **81** and **82** also showed inhibition against tumour cell lines (human prostate PC3, colorectal LoVo, colorectal adenocarcinoma CACO-2, and breast MDA-468 cancer cell lines) with IC_50_ values ranging from 2.9–9.5 µg/mL [38].

Other compounds isolated from Sulawesi included nakijiquinone V (**83**), isolated from *Dactylospongia elegans* from Tahuna, Sangihe Islands, North Sulawesi [39]. It has three methyl groups attached to a decalin system with an exocyclic double bond [39] (Figure 6). Another was a new meroditerpene—namely, Halioxepine (**84**), which was isolated from an Indonesian marine sponge from the genus *Haliclona* collected in Baubau, Southeast Sulawesi [40]. This compound showed moderate cytotoxicity against the rat bladder tumour NBT-T2 cells with IC_50_ of 11.6 µM and antioxidant activity against 1,1-diphenyl-2-picrylhydrazyl (DPPH) with IC_50_ of 7.7 µM [40]. Recently, two new terpenoids, melophluosides A (**85**) and B (**86**), were successfully isolated from *Melophlus sarassinorum* collected in Siladen, North Sulawesi. These new compounds belong to the triterpene galactosides of the pouoside class, with compound **85** as the first without an oxygenated group at C-11. Both compounds exhibited moderate cytotoxic activity against HeLa cells with IC50 values of 11.6 and 9.7 mM, respectively [41].

Two new cycoldepsipeptide jaspamide derivatives, jaspamide Q (**87**) and jaspamide R (**88**), were identified from the marine sponge *J. splendens* collected in neighbouring islands near East Kalimantan [42]. They exhibited potent cytotoxic activity against the mouse lymphoma L5187Y cancer cell line with IC_50_ values of <126.8 µM and <203 µM, respectively [42].

Novel tridecapeptides of the theonellapeptolide family were isolated from *T. swinhoei* collected from Bunaken Marine Park in Manado, North Sulawesi—namely, sulfinyltheonellapeptolide (**89**) and theonellapeptolide If (**90**) [43]. These compounds differ in the abrine moiety. Compound **89** is a theonellapeptolide with a methylsulfinylacetyl group at the *N*-terminus, while compound **90** was the first theonellapeptolide with four valine residues (Figure 7). Both compounds showed significant antiproliferative activity against human liver HepG2 cancer cells with the same IC_50_ value of 3 µM [43].

*S. mirabilis* is also one of the rich sources of diverse secondary metabolites, as exemplified by the isolation of six new depsipeptides with two different structural classes named celebesides A-C (**91–93**) from the species collected in Sulawesi [44]. Compounds **91–93** are cyclic depsipeptides with a polyketide moiety and five amino acid residues (Figure 7). The celebesides A and B (**91, 92**) possess a 3-carbamoyl threonine and a phosphoserine residue, which is quite uncommon. *S. mirabilis* also produced the peptides theopapuamides B–D (**94**–**96**), which are undecapeptides with an *N*-terminal fatty acid moiety, with theopapuamide D (**96**) containing a rare homoisoleucine residue (Figure 7). These compounds exhibited several bioactivities. Theopapuamides B and C (**94, 95**) were found to have relatively strong antifungal activity against amphotericin B-resistant *C. albicans* [44]. Additionally, celebeside A **(91)** and theopapuamide B **(94)** neutralized HIV-1 in a single-round infectivity assay with an IC_50_ value of 2.1 ± 0.4 µM and 499.7 ± 0.3 µg/mL [44]. Celebeside **91** and theopapuamides **94** and **95** also demonstrated cytotoxicity against human colon carcinoma cells with IC_50_ values 9.9 µM, 1.3 µM, and 2.5 µM, respectively [44]. Although they were potent against human colon HCT116 carcinoma cells, celebesides A, B, and C **(91-93)** were also found to be cytotoxic for healthy cell lines [44].

Haloirciniamide A (**97**) and seribunamide A (**98**), new polyhalogenated peptides, have been isolated from *Ircinia* sp. collected from the coast of Thousand Islands. Compound **97** was the first dibromopyrrole cyclopeptide with a chlorohistidine ring, while compound **98** possess a rare tribromopyrrole ring. Unfortunately, both compounds did not show significant cytotoxicity against four human tumour cell lines [45].

Four new endoperoxyketal polyketides, manadoperoxides A-D (**99**–**102**), were isolated from the sponge *Plakortis* cfr. *simplex* obtained from the Bunaken Marine Park of Manado, North Sulawesi [46]. In these compounds, the methoxy group at C-6 is replaced by either a methyl or ethyl group instead of a peroxyketal-type (Figure 8), making them slightly different from those previously isolated from the same species in the Caribbean. All compounds isolated from that sponge *Plakortis* cfr. *simplex* showed moderate antimalarial activity against D10 and W2 strains of *Plasmodium falciparum* [46].

A novel cytotoxic macrolide, namely Callyspongiolide (**103**), was isolated from the Indonesian marine sponge *Calyspongia* sp [47]. This compound has a notable feature of a conjugated diene-ynic sidechain ending at a benzene ring with bromine, which was not found in the previously isolated marine macrolides (Figure 9). Its cytotoxicity against human acute T cell leukaemia Jurkat J16 T and human Burkitt’s lymphoma Ramos B lymphocytes showed remarkable IC_50_ values of 70 and 60 nM, respectively, as well as EC_50_ values of 80 and 50 nM, respectively [47].

A novel A-nor steroid, namely clathruhoate (**104**), was isolated from a marine sponge *Clathria* sp., collected from Bintang Samudra Marine Education Park, Southeast Sulawesi [48]. Moreover, a new nortriterpenoid saponin, namely sarasinoside S (**105**), was identified from the marine sponge *Petrosia* sp. obtained in North Sulawesi [49]. Compound **105** was the first among the nortriterpenes in the saranoside family with a degraded side chain (Figure 9).

### 2.2. Tunicates

Tryptamine-derived alkaloids named leptoclinidamide (**106**) and (-)-leptoclinidamine B (**107**) were isolated from tunicate *L. dubius* collected from Lembeh Strait, North Sulawesi [50]. Compound **107** is the enantiomer of the previously isolated compound, (+)-leptoclinidamine B. These compounds have a unique amide moiety with two β-alanine units and d-arginine moiety, respectively [50].

Eleven compounds were identified from tunicate *Lissoclinum* cf. *badium* collected in Manado, Indonesia. Among these 11 compounds, there were four new polysulfur aromatic alkaloids named lissoclibadins 4–7 (**108**–**111**) (Figure 10), isolated with seven known alkaloids called lissoclibadins 1–3, lissoclinotoxins E and F, 3,4-dimethoxy6-(2′-*N*,*N*-dimethylaminoethyl)-5-(methylthio)benzotrithiane, and *N*,*N*-dimethyl-5-(methylthio)varacin [51]. Bioactivities of these compounds included inhibition against yeast *Saccharomyces cerevisiae* demonstrated by compounds **109**–**111**, and antimicrobial activity against *S. aureus* and *E. coli* evident from compounds **108**–**111** [51].

Other alkaloids polycarpathiamines A and B (**112** and **113**), were isolated from tunicate *Polycarpa aurata* (Table 2). Both alkaloid compounds have a rare 1,2,4-thiadiazole ring (Figure 10). Compound **112** demonstrated potent cytotoxic activity against mouse lymphoma L5187Y cancer cell line with an IC_50_ value of 0.41 µM [52].

Another two novel alkaloids have recently been isolated from another *P. aurata from* the coast of Siladen, North Sulawesi [53]. Featuring a *p*-methoxyphenyl group, these compounds were named polyaurines A (**114**) and B (**115**). Compound **114** deformed eggs of parasite *Schistosoma mansoni*, yet it was not toxic to mammalian cells [53]. Two new benzoyl derivatives from this species, ethyl 2-(4-methoxyphenyl)-2-oxoacetate (**116**) and methyl 2-(4-hydroxyphenyl)-2-oxoacetate (**117**), were also isolated (Figure 11).

Two new cyclic hexapeptides, mollamides B (**118**) and C (**119**) (Figure 11), were isolated from *D. mole* found in Manado Bay, Indonesia. Compound **118** was found to have an antiparasitic activity against *Plasmodium falciparum* D6 clone (IC_50_ = 2.9 µM), *P. falciparum* W2 clone (IC50 = 3 µM), and *Leishmania donovani* (IC_50_ = 25.9 µM and IC_90_ = 50.3 µM) [54]. Compound **118** also displayed antiviral activity against HIV-1 in human PBM cells with an EC_50_ value of 48.7 µM and anticancer activity against the non-small lung H460, the breast MCF-7, and the human glioblastoma CNS SF268 cancer cell lines [54]. Furthermore, compound **119** showed anticancer activity with a unit zone differential value of 100 against mouse lymphocytic leukaemia L1210, human colon HCT116, and human lung H125 cells. This compound also showed a differential value of 250 against murine colon 38 [54]. However, compounds **118** and **119** did not show anti-inflammatory activity either in vitro in a cell-based assay through a cyclooxygenase enzyme (COX-2) activity assay or in vivo in rat neonatal microglia [54]. Furthermore, compounds **118** and **119** did not show antimicrobial activity against MRSA, *Mycobacterium intracellulaire, Candida albicans, C. glabrata, C. krusei*, nor *Cryptococcus neoformans* [54].

### 2.3. Soft Corals

Terpenoids were often isolated from soft corals, with varying degrees of bioactivities (Table 3). Sarcofuranocembrenolides A (**120**) and B (**121**) were isolated from the soft coral *Sarcophyton* sp. collected in North Sulawesi [55]. Cembranoid **120** is a bisnorcembrenolide featuring a unique carbon skeleton of 8,19-bisnorfuranocembrenolide (Figure 12). On the other hand, cembranoid **121** is a furanocembrenolide with a C_1_ unit (C-20) attached to C-10. In the ordinary cembrenolides, the C_1_ unit is attached to C-12 [55].

Three C-4 norcembranoids-type macrocyclic diterpenoids, namely chloroscabrolide A (**122**), chloroscabrolide B (**123**), and prescarbolide (**124**), were isolated from *Sinularia* sp. collected from Bunaken Marine Park, Manado, North Sulawesi [56]. Compounds **122** and **123** are two of very few chlorinated compounds from soft coral metabolites and the second example within the class of cembranoids (Figure 12). These two compounds also feature an oxygen bridge connecting C-13 and C-15, which is quite unusual. Meanwhile, the terpenoid prescarbolide (**124**) is believed to be the precursor of the scabrolide/leptocladolide family of cembranoids [56].

Two new isomeric eunicellin-type diterpenoids were isolated from an Indonesian octocoral *Cladiella* sp. in the West Pacific Ocean [57]. Cladielloide A (**125**) and cladielloide B (**126**) both possess a 2-hydroxybutyroxy group in their structures (Figure 13). Compound **126** exhibited a potent inhibitory effect on superoxide anion generation and elastase release by human neutrophils at 22 µM. It also showed moderate cytotoxicity against the acute lymphocytic leukaemia CCRF-CEM tumour cells with an IC_50_ value of 10.1 µM [57].

A terpenoid compound, 3,4-epoxy-nephtenol (**127**), was isolated from the soft coral *Nephthea* sp. found in Seribu Islands, DKI Jakarta [58]. Compound **127** showed weak inhibitory growth against three human tumour cell lines, i.e., human glioblastoma SF268, human breast MCF-7, and human non-small cell lung H460 cancer cells [58].

The latest finding on terpenoids from soft coral was discovered from *Anthelia* sp. collected at Banten, West Java, which successfully isolated two new dolabellane diterpenoids, sangiangol A (**128**) and sangiangol B (**129**). These two compounds were found to show weak toxicity against NBT-T2 rat bladder epithelial cells (BRC-1370) at 18 and 28.2 µM, respectively [59]. The first reported zoanthamine-type alkaloid from a marine invertebrate other than zoanthids was named lobozoanthamine (**130**), isolated from the Indonesian soft coral *Lobophytum* sp. collected from the Bunaken Marine Park, Manado, North Sulawesi [60]. However, there is no report on the bioactivity of the compounds **120**–**125** and **130** (Table 3).

## 3. Conclusions

Indonesian marine biodiversity holds immense potential for drug discovery and bioprospecting, supporting the continuous exploration and investigation of Indonesian MNPs from marine invertebrates such as sponges, tunicates, and soft corals. This is further justified by no less than a hundred and thirty novel compounds reported in this review from 43 publications throughout 2007–2020, the majority of which were isolated from the Indonesian marine invertebrates collected from the North region of Sulawesi, Indonesia (Figure 13).

Indonesian sponges have been known as a major source of many novel MNPs. In this review, sponges were found to contribute to 105 novel compounds reported between 2007–2020, the majority being alkaloids (Figure 14). Most of the sponge alkaloids were isolated from the genera *Aplysinella* and *Acanthostrongulophora*. The next large groups of metabolites isolated from Indonesian sponges were terpenes and peptides. Nearly all the sponge terpenoids were isolated from the order Dictyoceratida, in which the dominant genera were *Lamellodysidea* and *Carteriospongia*.

Meanwhile, all the peptides, with the exception of two peptides from *Ircinia* sp., were isolated from the order Tetractinellida. On the other hand, macrolides, steroids, and saponins were the least frequently isolated metabolites from Indonesian sponges. The sponge *Plakortis simplex* was the sole contributor of the polyketides isolated from Indonesia (Table 1).

During 2007–2020, ten alkaloids were reported from Indonesian tunicates, mainly from the family *Didemnidae* and *Styelidae*. Soft corals are also a good source of novel metabolites and are well-known as a producer of terpenoids, particularly the group of terpenes and cembranoid diterpenes. Soft coral-derived terpenoids have received significant attention in Indonesian MNPs research (Table 3). Compared to other invertebrates with alkaloids as the most abundant secondary metabolites, only one alkaloid was found in Indonesian soft coral species in this review.

Indonesian soft corals and tunicates yielded far lower novel secondary metabolites than sponges reported throughout 2007–2020. Therefore, this review highlighted the opportunity to explore further the chemical diversity and the biological activity of tunicates from Indonesian waters.

Diverse biological activities were shown by fifty per cent of the compounds isolated from the three Indonesian marine invertebrates discussed (Figure 14). The majority of compounds in this review were reported to possess cytotoxic or antiproliferative activity against various cancer cell lines. Four of these compounds showed a remarkable cytotoxic activity with IC_50_ values of less than 1 µM, namely compounds **30, 38, 103** and **112**. Furthermore, four compounds showed IC_50_ values between 1–4 µM, and another eight showed moderate cytotoxic with an IC_50_ value of less than 10 µM. Most of these compounds belong to the alkaloid group, followed by peptides and terpenoids. It is, therefore, evident that the exploration of potential anticancer drugs from Indonesian marine resources warrants further investigation.

The bioactivity of compounds isolated from the Indonesian marine invertebrates (sponges, tunicates, and soft corals) as antimicrobial, antifungal, or antiviral was also described in this review. Unfortunately, little or no such activities were yet to be found from many compounds derived from the three Indonesian marine invertebrates, demonstrating the gap in the knowledge of their beneficial biological activity other than their cytotoxic activity. This knowledge gap opens up the opportunity for further research focusing on exploring and harnessing the potential of Indonesian marine invertebrates as sources of compounds with antimicrobial, antifungal, or antiviral activities, as well as further exploration of Indonesian marine biodiversity for the discovery of novel bioactive compounds.

## Figures and Tables

**Figure 1 molecules-26-01898-f001:**
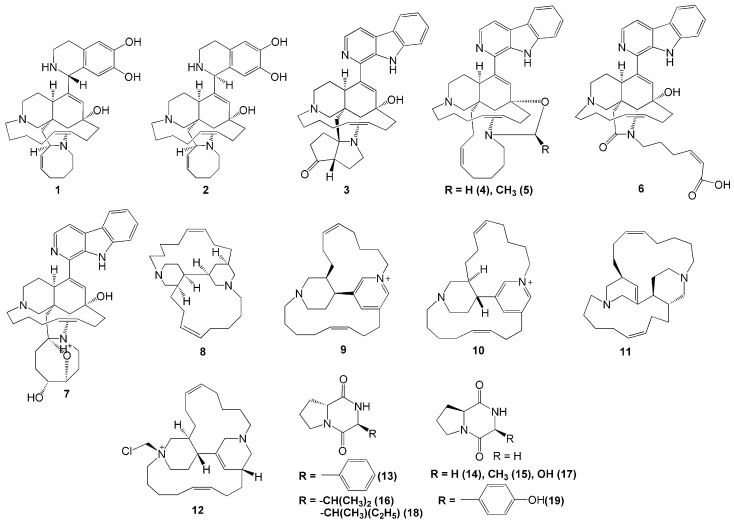
Chemical structures of **1**–**19.**

**Figure 2 molecules-26-01898-f002:**
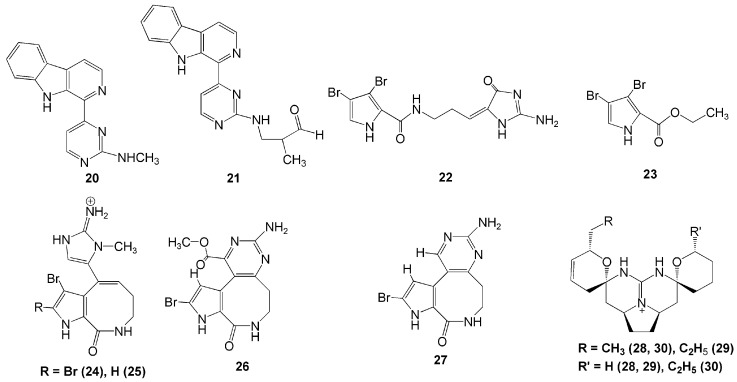
Chemical structures of **20–30.**

**Figure 3 molecules-26-01898-f003:**
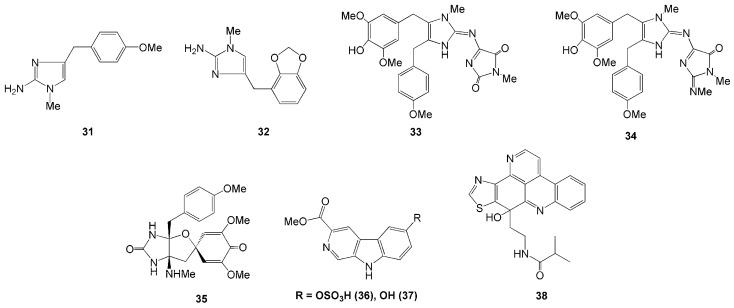
Chemical Structures of **31–38.**

**Figure 4 molecules-26-01898-f004:**
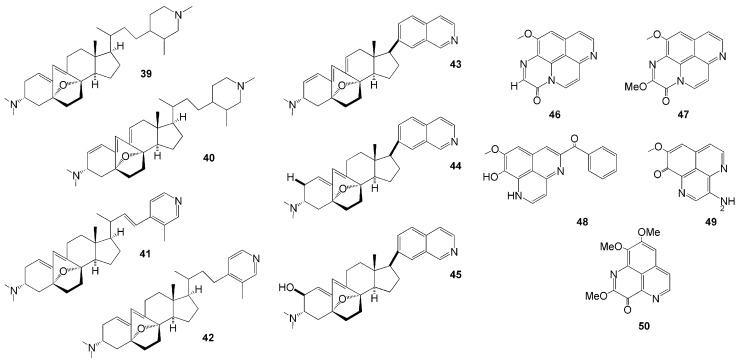
Chemical structures of **39–50.**

**Figure 5 molecules-26-01898-f005:**
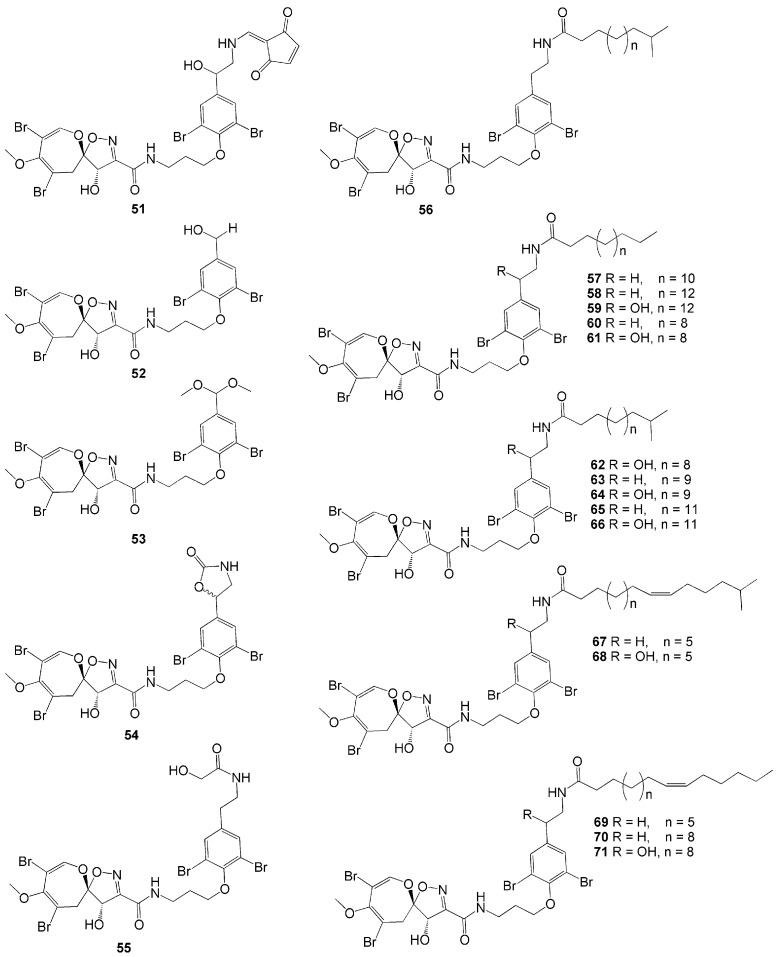
Chemical structures of **51**–**71.**

**Figure 6 molecules-26-01898-f006:**
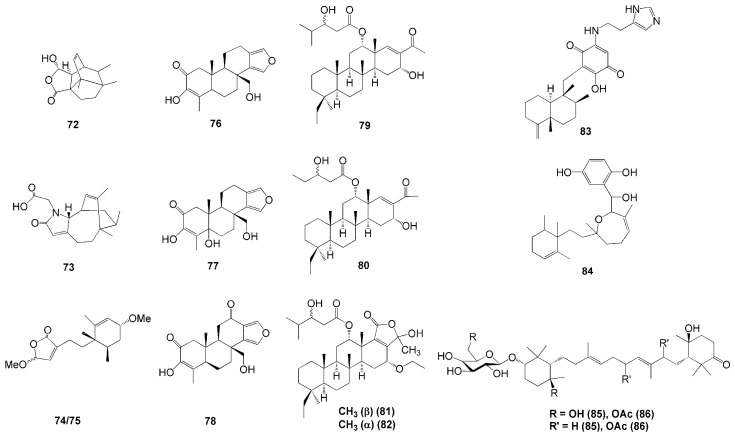
Chemical structures of **72–86.**

**Figure 7 molecules-26-01898-f007:**
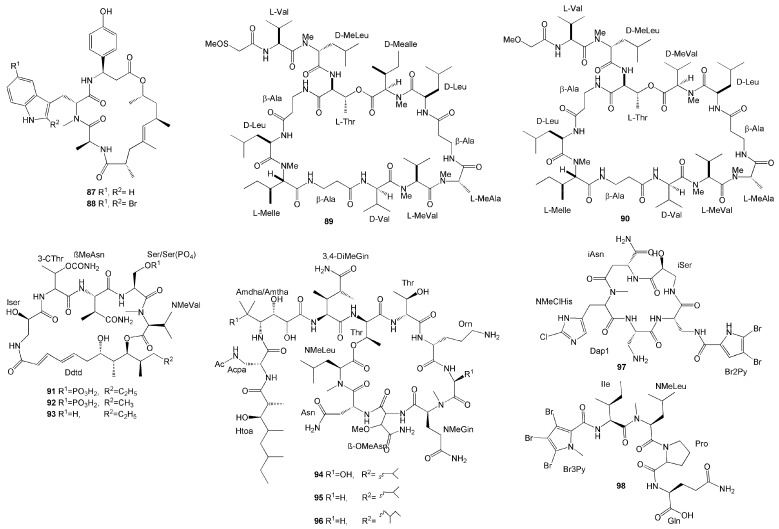
Chemical structures of **87–98.**

**Figure 8 molecules-26-01898-f008:**
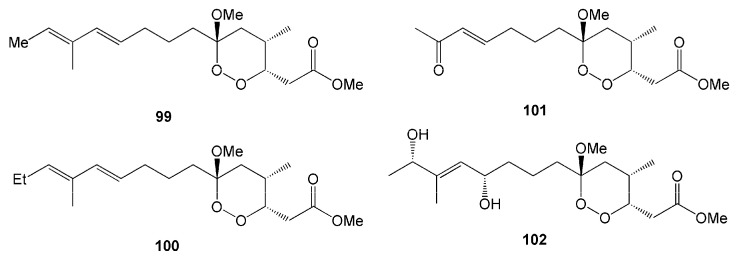
Chemical structures of **99**–**102.**

**Figure 9 molecules-26-01898-f009:**
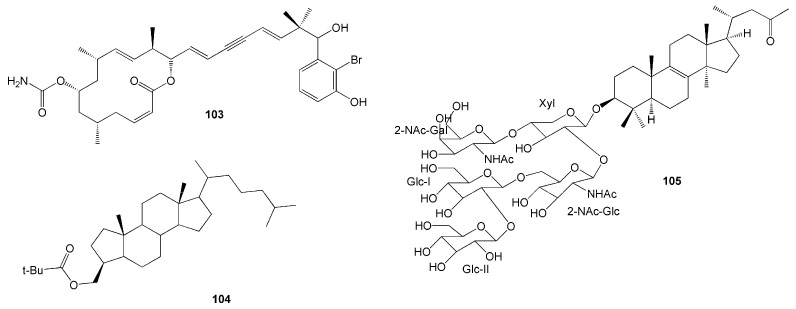
Chemical structures of **103**–**105.**

**Figure 10 molecules-26-01898-f010:**
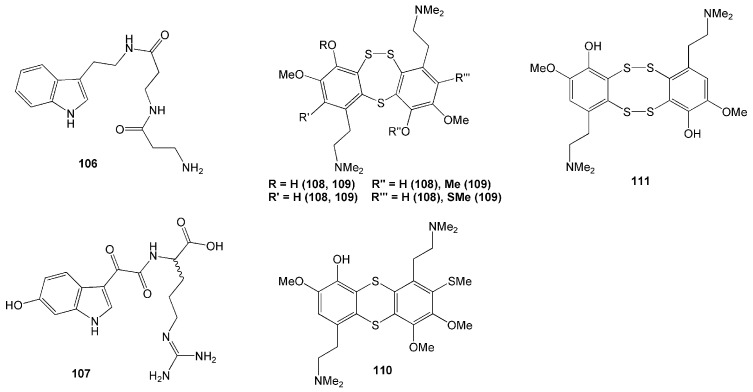
Chemical structures of **106**–**111.**

**Figure 11 molecules-26-01898-f011:**
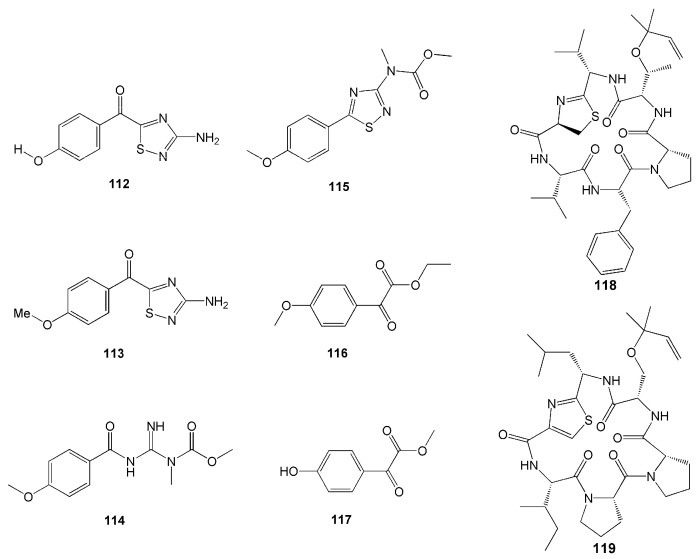
Chemical structures of **112**–**119.**

**Figure 12 molecules-26-01898-f012:**
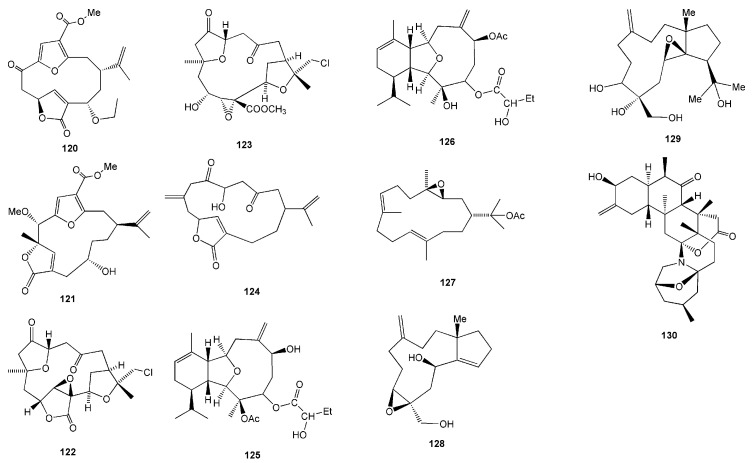
Chemical structures of **120**–**130.**

**Figure 13 molecules-26-01898-f013:**
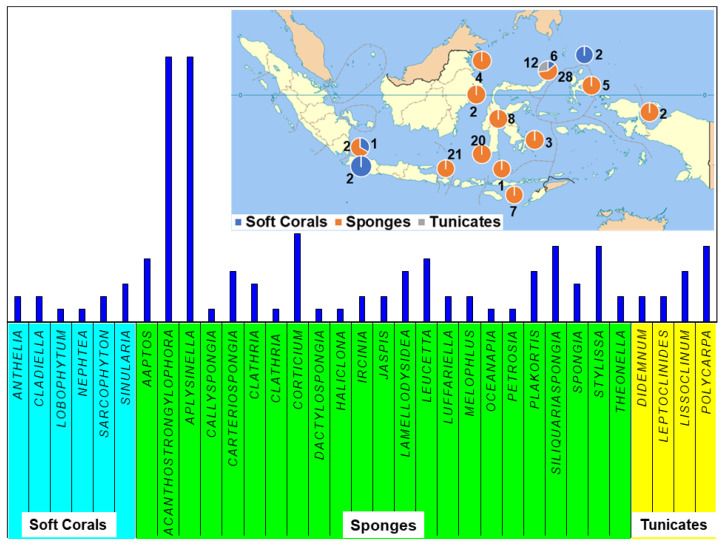
The distribution of sample origin and the division of compound class by genus.

**Figure 14 molecules-26-01898-f014:**
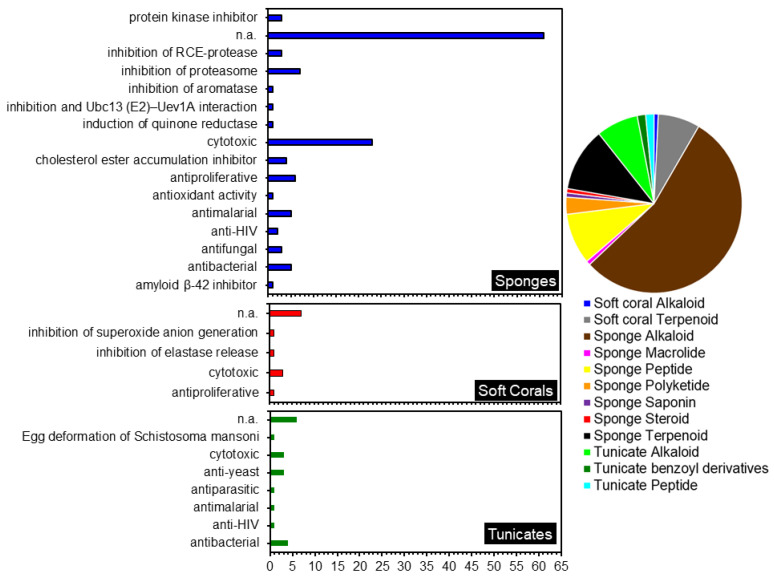
Biological activities of compounds isolated from Indonesian marine invertebrates (left) and the division of genus by compound class (right).

**Table 1 molecules-26-01898-t001:** Summary of the marine natural products isolated from the marine sponges from Indonesian oceans.

Compound	Compound Class	Species	Biological Activity	Ref
Acanthomanzamine A (**1**)	Alkaloid	*Acanthostrongylophora ingens*	Cytotoxic against human cervical HeLa cells; inhibition of proteasome; cholesterol ester accumulation inhibitor	[18]
Acanthomanzamine B (**2**)	Alkaloid	*A. ingens*	Cytotoxic against human cervical HeLa cells; inhibition of proteasome; cholesterol ester accumulation inhibitor	[18]
Acanthomanzamine C (**3**)	Alkaloid	*A. ingens*	n.a.	[18]
Acanthomanzamine D (**4**)	Alkaloid	*A. ingens*	Cytotoxic against human cervical HeLa cells; inhibition of proteasome; cholesterol ester accumulation inhibitor	[18]
Acanthomanzamine E (**5**)	Alkaloid	*A. ingens*	Cytotoxic against human cervical HeLa cells; inhibition of proteasome; cholesterol ester accumulation inhibitor	[18]
Acantholactam (**6**)	Alkaloid	*A. ingens*	Inhibition of proteasome	[19]
Pre-*neo*-kauluamine (**7**)	Alkaloid	*A. ingens*	Inhibition of proteasome	[19]
Acanthocyclamine A (**8**)	Alkaloid	*A. ingens*	Antimicrobial against *E. coli;* inhibitor of amyloid β-42 production	[20,21]
*Epi*-tetradehydrohalicyclamine B (**9**)	Alkaloid	*A. ingens*	n.a.	[21]
Tetradehydrohalicyclamine B (**10**)	Alkaloid	*A. ingens*	n.a.	[21]
Halicyclamine B (**11**)	Alkaloid	*A. ingens*	Antimicrobial against *S. aureus*	[21]
Chloromethylhalicyclamine B (**12**)	Alkaloid	*A. ingens*	Protein kinase CK1 δ/ε inhibitor	[21]
Cyclo (d-Pro-l-Phe) (**13**)	Alkaloid	*A. ingens*	Protein kinase CDK2/cyclin A inhibitor	[21]
Cyclo (l-Pro-Gly) (**14**)	Alkaloid	*A. ingens*	n.a.	[21]
Cyclo (l-Pro-l-Ala) (**15**)	Alkaloid	*A. ingens*	n.a.	[21]
Cyclo (d-Pro-l-Val) (**16**)	Alkaloid	*A. ingens*	n.a.	[21]
Cyclo (l-Pro-Ser) (**17**)	Alkaloid	*A. ingens*	n.a.	[21]
Cyclo (d-Pro-l-Ile) (**18**)	Alkaloid	*A. ingens*	n.a.	[21]
Cyclo (l-Pro-l-Tyr) (**19**)	Alkaloid	*A. ingens*	n.a.	[21]
Ingenine C (**20**)	Alkaloid	*A. ingens*	Cytotoxic against human breast MCF-7 and colorectal HCT116 cancer cells	[22]
Ingenine D (**21**)	Alkaloid	*A. ingens*	Cytotoxic against human breast MCF-7 and colorectal HCT116 cancer cells	[22]
Dispacamide E (**22**)	Alkaloid	*Stylissa massa*	Protein kinase inhibitor (GSK-3, DYRK1A, and CK-1)	[23]
Ethyl 3,4-dibromo-1*H*-pyrrole-2-carboxylate (**23**)	Alkaloid	*S. massa*	n.a.	[23]
12-*N*-methyl stevensine (**24**)	Alkaloid	*Stylissa* sp.	Cytotoxic against mouse lymphoma L5187Y cancer cell line	[24]
12-*N*-methyl-2-debromostevensine (**25**)	Alkaloid	*Stylissa* sp.	n.a.	[24]
3-debromolatonduine B methyl ester (**26**)	Alkaloid	*Stylissa* sp.	n.a.	[24]
3-debromolatonduine A (**27**)	Alkaloid	*Stylissa* sp.	n.a.	[24]
Crambescidin 345 (**28**)	Alkaloid	*Clathria bulbotoxa*	Cytotoxic against the human epidermal A431 carcinoma cell line	[25]
Crambescidin 361 (**29**)	Alkaloid	*C. bulbotoxa*	Cytotoxic against the human epidermal A431 carcinoma cell line	[25]
Crambescidin 373 (**30**)	Alkaloid	*C. bulbotoxa*	Cytotoxic against the human epidermal A431 carcinoma cell line	[25]
Methyldorimidazole (**31**)	Alkaloid	*Leucetta chagosensis*	n.a.	[26]
Preclathridine B (**32**)	Alkaloid	*L. chagosensis*	n.a.	[26]
Naamidine H (**33**)	Alkaloid	*L. chagosensis*	Cytotoxic against human cervical HeLa cells	[27,28]
Naamidine I (**34**)	Alkaloid	*L. chagosensis*	Cytotoxic against human cervical HeLa cells	[27]
Spironaamidine (**35**)	Alkaloid	*Leucetta microraphis*	Antibacterial activity against *B. cereus*	[28]
Variabine A (**36**)	Alkaloid	*Luffariella variabilis*	n.a.	[29]
Variabine B (**37**)	Alkaloid	*L. variabilis*	Inhibition of proteasome and Ubc13 (E2)–Uev1A interaction	[29]
Sagitol C (**38**)	Alkaloid	*Oceanapia* sp.	Cytotoxic against mouse lymphoma L5187Y, human cervical HeLa, and rat pheochromocytoma PC12 cells	[30]
Cortistatin E (**39**)	Alkaloid	*Corticium complex*	n.a.	[31]
Cortistatin F (**40**)	Alkaloid	*C. complex*	n.a.	[31]
Cortistatin G (**41**)	Alkaloid	*C. complex*	n.a.	[31]
Cortistatin H (**42**)	Alkaloid	*C. complex*	n.a.	[31]
Cortistatin J (**43**)	Alkaloid	*C. complex*	Cytostatic antiproliferative activity against human umbilical vein endothelial cells (HUVECs)	[32]
Cortistatin K (**44**)	Alkaloid	*C. complex*	n.a.	[32]
Cortistatin L (**45**)	Alkaloid	*C. complex*	n.a.	[32]
11-Methoxy-3*H*-[1,6]naphthyridino[6,5,4-*def*]quinoxalin-3-one (**46**)	Alkaloid	*Aaptos suberitoides*	n.a.	[33]
2,11-Dimethoxy-3*H*-[1,6]naphthyridino[6,5,4-*def*]quinoxalin-3-one (**47**)	Alkaloid	*A. suberitoides*	n.a.	[33]
5-Benzoydemethylaaptamine (**48**)	Alkaloid	*A. suberitoides*	Cytotoxic against mouse lymphoma L5187Y cancer cell line	[33]
3-Aminodemethyl(oxy)aaptamine (**49**)	Alkaloid	*A. suberitoides*	n.a.	[33]
2-methoxy-3-oxoaaptamine (**50**)	Alkaloid	*Aaptos* sp.	Antibacterial against *Mycobacterium smegmatis*	[34]
19-Hydroxypsammaplysin E (**51**)	Alkaloid	*Aplysinella strongylata*	Antimalarial against *P. falciparum*	[35]
Psammaplysin K (**52**)	Alkaloid	*A. strongylata*	n.a.	[35]
Psammaplysin K dimethoxy acetal (**53**)	Alkaloid	*A. strongylata*	n.a.	[35]
Psammaplysin L (**54**)	Alkaloid	*A. strongylata*	n.a.	[35]
Psammaplysin M (**55**)	Alkaloid	*A. strongylata*	n.a.	[35]
Psammaplysin N (**56**)	Alkaloid	*A. strongylata*	n.a.	[35]
Psammaplysin O (**57**)	Alkaloid	*A. strongylata*	n.a.	[35]
Psammaplysin P (**58**)	Alkaloid	*A. strongylata*	n.a.	[35]
19-Hydroxypsammaplysin P (**59**)	Alkaloid	*A. strongylata*	n.a.	[35]
Psammaplysin Q (**60**)	Alkaloid	*A. strongylata*	n.a.	[35]
19-Hydroxypsammaplysin Q (**61**)	Alkaloid	*A. strongylata*	n.a.	[35]
Psammaplysin R (**62**)	Alkaloid	*A. strongylata*	n.a.	[35]
Psammaplysin S (**63**)	Alkaloid	*A. strongylata*	n.a.	[35]
19-Hydroxypsammaplysin S (**64**)	Alkaloid	*A. strongylata*	n.a.	[35]
Psammaplysin T (**65**)	Alkaloid	*A. strongylata*	n.a.	[35]
19-Hydroxypsammaplysin T (**66**)	Alkaloid	*A. strongylata*	n.a.	[35]
Psammaplysin U (**67**)	Alkaloid	*A. strongylata*	n.a.	[35]
19-Hydroxypsammaplysin U (**68**)	Alkaloid	*A. strongylata*	n.a.	[35]
Psammaplysin V (**69**)	Alkaloid	*A. strongylata*	n.a.	[35]
Psammaplysin W (**70**)	Alkaloid	*A. strongylata*	n.a.	[35]
19-Hydroxypsammaplysin W (**71**)	Alkaloid	*A. strongylata*	n.a.	[35]
Lamellodysidine A (**72**)	Terpenoid	*Lamellodysidea herbacea*	n.a.	[36]
Lamellodysidine B (**73**)	Terpenoid	*L. herbacea*	n.a.	[36]
*O,O*-dimethyllingshuiolide A (**74**)	Terpenoid	*L. herbacea*	n.a.	[36]
11-*epi*-*O,O*-dimethyllingshuiolide A (**75**)	Terpenoid	*L. herbacea*	n.a.	[36]
18-nor-3,17-dihydroxyspongia-3,13(16),14-trien-2-one (**76**)	Terpenoid	*Spongia* sp.	Inhibition of aromatase	[37]
18-nor-3,5,17-trihydroxyspongia-3,13(16),14-trien-2-one (**77**)	Terpenoid	*Spongia* sp.	n.a.	[37]
Spongiapyridine (**78**)	Terpenoid	*Spongia* sp.	n.a.	[37]
20,24-bishomo-25-norscalarane 1 (**79**)	Terpenoid	*Carteriospongia foliascens*	Antiproliferative activity against human prostate PC3, colorectal LoVo, colorectal CACO-2, and breast MDA-468 cancer cells; inhibition of RCE-protease	[38]
20,24-bishomo-25-norscalarane 2 (**80**)	Terpenoid	*C. foliascens*	n.a.	[38]
20,24-bishomoscalarane ketals 3 (**81**)	Terpenoid	*C. foliascens*	Antiproliferative activity against human prostate PC3, colorectal LoVo, colorectal CACO-2, and breast MDA-468 cancer cells; inhibition of RCE-protease	[38]
20,24-bishomoscalarane ketals 4 (**82**)	Terpenoid	*C. foliascens*	Antiproliferative activity against human prostate PC3, colorectal LoVo, colorectal CACO-2, and breast MDA-468 cancer cells; inhibition of RCE-protease	[38]
nakijiquinone V (**83**)	Terpenoid	*Dactylospongia elegans*	n.a.	[39]
Halioxepine (**84**)	Terpenoid	*Haliclona* sp.	Cytotoxic against rat bladder tumour NBT-T2; antioxidant activity	[40]
Melophluoside A (**85**)	Terpenoid	*Melophlus sarasinorum*	Cytotoxic against human cervical HeLa cells	[41]
Melophluoside B (**86**)	Terpenoid	*M. sarasinorum*	Cytotoxic against human cervical HeLa cells	[41]
Jaspamide Q (**87**)	Peptide	*Jaspis splendens*	Cytotoxic against mouse lymphoma L5187Y cancer cell line	[42]
Jaspamide R (**88**)	Peptide	*J. splendens*	Cytotoxic against mouse lymphoma L5187Y cancer cell line	[42]
Sulfinyltheonellapeptolide (**89**)	Peptide	*Theonella swinhoei*	Antiproliferative activity against human liver HepG2 cancer cell line	[43]
Theonellapeptolide If (**90**)	Peptide	*T. swinhoei*	Antiproliferative activity against human liver HepG2 cancer cell line	[43]
Celebeside A (**91**)	Peptide	*Siliquariaspongia mirabilis*	Cytotoxic against HCT116; anti-HIV	[44]
Celebeside B (**92**)	Peptide	*S. mirabilis*	n.a.	[44]
Celebeside C (**93**)	Peptide	*S. mirabilis*	n.a.	[44]
Theopapuamide B (**94**)	Peptide	*S. mirabilis*	Cytotoxic against human colorectal HCT116 cancer cell line; anti-HIV	[44]
Theopapuamide C (**95**)	Peptide	*S. mirabilis*	Cytotoxic against human colorectal HCT116 cancer cell line	[44]
Theopapuamide D (**96**)	Peptide	*S. mirabilis*	n.a.	[44]
Haloirciniamide A (**97**)	Peptide	*Ircinia* sp.	n.a.	[45]
Seribunamide A (**98**)	Peptide	*Ircinia* sp.	n.a.	[45]
Manadoperoxide A (**99**)	Polyketide	*Plakortis* cfr. *simplex*	Antimalarial against *P. falciparum*	[46]
Manadoperoxide B (**100**)	Polyketide	*Plakortis* cfr. *simplex*	Antimalarial against *P. falciparum*	[46]
Manadoperoxide C (**101**)	Polyketide	*Plakortis* cfr. *simplex*	Antimalarial against *P. falciparum*	[46]
Manadoperoxide D (**102**)	Polyketide	*Plakortis* cfr. *simplex*	Antimalarial against *P. falciparum*	[46]
Callyspongiolide (**103**)	Macrolide	*Callyspongia* sp.	Cytotoxic against Jurkat J16 T and Ramos B lymphocytes	[47]
Clathruhoate (**104**)	Steroid	*Clathria* sp.	n.a.	[48]
Saranoside S (**105**)	Saponin	*Petrosia* sp.	n.a.	[49]

**Table 2 molecules-26-01898-t002:** Summary of the marine natural products isolated from the marine tunicates from Indonesian oceans.

Compound	Compound Class	Species	Biological Activity	Ref
Leptoclinidamide (**106**)	Alkaloid	*Leptoclinides dubius*	n.a.	[50]
(-)-leptoclinidamine B (**107**)	Alkaloid	*L. dubius*	n.a.	[50]
Lissoclibadin 4 (**108**)	Alkaloid	*Lissoclinum* cf. *badium*	Antibacterial against *S. aureus* and *E. coli*	[51]
Lissoclibadin 5 (**109**)	Alkaloid	*Lissoclinum* cf. *badium*	Antibacterial against *S. aureus* and *E. coli;* anti-yeast activity against *S. cerevisiae*	[51]
Lissoclibadin 6 (**110**)	Alkaloid	*Lissoclinum* cf. *badium*	Antibacterial against *S. aureus* and *E. coli;* anti-yeast activity against *S. cerevisiae*	[51]
Lissoclibadin 7 (**111**)	Alkaloid	*Lissoclinum* cf. *badium*	Antibacterial against *S. aureus* and *E. coli*; anti-yeast activity against *S. cerevisiae*	[51]
Polycarpathiamine A (**112**)	Alkaloid	*Polycarpa aurata*	Cytotoxic against mouse lymphoma L5187Y cancer cell line	[52,53]
Polycarpathiamine B (**113**)	Alkaloid	*P. aurata*	n.a.	[52]
Polyaurine A (**114**)	Alkaloid	*P. aurata*	Egg deformation of *Schistosoma mansoni*	[53]
Polyaurine B (**115**)	Alkaloid	*P. aurata*	n.a.	[53]
Ethyl 2-(4-methoxyphenyl)-2-oxoacetate (**116**)	benzoyl derivatives	*P. aurata*	n.a	[53]
Methyl 2-(4-hydroxyphenyl)-2-oxoacetate (**117**)	benzoyl derivatives	*P. aurata*	n.a.	[53]
Mollamide B (**118**)	Peptide	*Didemnum molle*	Cytotoxic against human non-small cell lung H460, breast MCF-7, and glioblastoma SF268 cells; antimalarial against *P. falciparum*; antiparasitic against Leishmania donovani; antiviral against HIV-1	[54]
Mollamide C (**119**)	Peptide	*D. molle*	Cytotoxic against mouse lymphocytic leukaemia L1210, human colorectal HCT116, lung H125, and murine colon MC 38 cancer cells	[54]

**Table 3 molecules-26-01898-t003:** Summary of the marine natural products isolated from the marine soft corals from Indonesian oceans.

Compound	Compound Class	Species	Biological Activity	Ref
Sarcofuranocembrenolide A (**120**)	Terpenoid	*Sarcophyton* sp.	n.a.	[55]
Sarcofuranocembrenolide B (**121**)	Terpenoid	*Sarcophyton* sp.	n.a.	[55]
Chloroscabrolide A (**122**)	Terpenoid	*Sinularia* sp.	n.a.	[56]
Chloroscabrolide B (**123**)	Terpenoid	*Sinularia* sp.	n.a.	[56]
Prescabrolide (**124**)	Terpenoid	*Sinularia* sp.	n.a.	[56]
Cladielloide A (**125**)	Terpenoid	*Cladiella* sp.	n.a.	[57]
Cladielloide B (**126**)	Terpenoid	*Cladiella* sp.	Cytotoxic against lymphocytic leukaemia CCRF-CEM cells; inhibition of superoxide anion generation; inhibition of elastase release	[57]
3,4-epoxy-nephthenol acetate (**127**)	Terpenoid	*Nephthea* sp.	Antiproliferative activity against human glioblastoma SF268, breast MCF-7, and non-small cell lung H460 cancer cells	[58]
Sangiangol A (**128**)	Terpenoid	*Anthelia* sp.	Cytotoxic rat bladder tumour NBT-T2 cell line	[59]
Sangiangol B (**129**)	Terpenoid	*Anthelia* sp.	Cytotoxic rat bladder tumour NBT-T2 cell line	[59]
Lobozoanthamine (**130**)	Alkaloid	*Lobophytum* sp.	n.a.	[60]

## Data Availability

Not Applicable.

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
