# Peer review of "Chemical Diversity and Biological Activity of Secondary Metabolites Isolated from Indonesian Marine Invertebrates"

_molecules, 2021, doi:10.3390/molecules26071898_

Round 1

Reviewer 1 Report

Dear Authors,

This review manuscript gathered all the published papers from 2007-2019 with the aim to give a summary of marine natural products isolated from invertebrates collected at Indonesia.

1) For this review paper, I think Authors should consider to cover additional one more year (2020), that is from 2007 to 2020. Because this review paper was submitted in 2021.

2) At lines 25-26, "NPs with vast chemical structures and potential pharmacological activities" are cited with very old references. Authors might consider below MDPI references:

a) Terpenoids from marine soft coral of the genus Xenia in 1977 to 2019. Molecules, 2020, 25, 5386. DOI: 10.3390/molecules25225386

b) Natural products from octocorals of the genus Dendronephthya (family Nephtheidae). Molecules, 2020, 25, 5957. DOI: 10.3390/molecules25245957

c) Chemical diversity and biological activities of marine sponges of the genus Suberea: a systematic review. Mar. Drugs, 2019, 17, 115. DOI: 10.3390/md17020115

d) Marine natural products. Nat. Prod. Rep., 2020, 37, 175. DOI: 10.1039/C9NP00069K

3) At lines 26-27, I have difficulty to understand this sentence "During 2012-2017 ... marine invertebrates,". Lines 26-30, should rewrite into a more focus fact, where the discovery of marine natural products from invertebrates has been increasing.

4) No. 110-111 from Table 1, mollamides B-C might be categorized as peptide instead of cyclic hexapeptides. 

5) Table 1, there are a lot of mistake when mentioned the compound name, for example "mollamide B and mollamide C" or "mollamides B and C". Another example is "chloroscabrolide A" not "chloroscabrolides A".

6) Figure 7, compound 85 and 86 were not drew properly, please redraw it. The dashed line was reversed and OH bond bended. 

7) Figure 7, compound 87 and 88 were not drew properly, the dashed line of Leu from the branch of cyclic peptide has reversed. Second, why only D-Me Val and D-Val were labelled? But amino acid residues were labeled in 89-94, Author have to consider consistency.

8) Figure 99, please enlarge your chemdraw template, as you can see the NH2 has squeezed in 99.

9) Figure 13 (B) and (C) are difficult to see, because small and blurred.

Author Response

Dear editors and reviewers,

Below are our responses to the comments of reviewers. We have copied and pasted their comments. The reviewers’ comments appear in italics and follow in the same order as written by the referee. Our responses appear in regular font.

Reviewer #1:

General comments

This review manuscript gathered all the published papers from 2007-2019 with the aim to give a summary of marine natural products isolated from invertebrates collected at Indonesia.

Specific comments

  1. For this review paper, I think Authors should consider to cover additional one more year (2020), that is from 2007 to 2020. Because this review paper was submitted in 2021.

Response:

Thank you very much for your suggestions. We have added the following recent references in the discussion of our revised manuscript:

  • Sadahiro, Y.; Hitora, Y.; Fukumoto, A.; Ise, Y.; Angkouw, E. D.; Mangindaan, R. E. P.; Tsukamoto, S. Melophluosides A and B, New Triterpene Galactosides from the Marine Sponge Melophlus sarasinorum. Tetrahedron Lett. 2020, 61 (20), 151852. https://doi.org/10.1016/j.tetlet.2020.151852.
  • Fernández, R.; Bayu, A.; Hadi, T. A.; Bueno, S.; Pérez, M.; Cuevas, C.; Putra, M. Y. Unique Polyhalogenated Peptides from the Marine Sponge Ircinia Mar. Drugs 2020, 18 (8), 1–10. https://doi.org/10.3390/MD18080396.
  • Hanif, N.; Murni, A.; Tanaka, J. Sangiangols A and B, Two New Dolabellanes from an Indonesian Marine Soft Coral, Anthelia Molecules 2020, 25 (17). https://doi.org/10.3390/molecules25173803.
  1. At lines 25-26, "NPs with vast chemical structures and potential pharmacological activities" are cited with very old references. Authors might consider below MDPI references:
  2. a) Terpenoids from marine soft coral of the genus Xenia in 1977 to 2019. Molecules, 2020, 25, 5386. DOI: 10.3390/molecules25225386
  3. b) Natural products from octocorals of the genus Dendronephthya (family Nephtheidae). Molecules, 2020, 25, 5957. DOI: 10.3390/molecules25245957
  4. c) Chemical diversity and biological activities of marine sponges of the genus Suberea: a systematic review. Mar. Drugs, 2019, 17, 115. DOI: 10.3390/md17020115
  5. d) Marine natural products. Nat. Prod. Rep., 2020, 37, 175. DOI: 10.1039/C9NP00069K

Response:

Thank you very much for your remarks. The references (c, d) have been cited in the article at lines 27-31. We have added the references (a, b) in the revised manuscript at lines 31-33).

  1. At lines 26-27, I have difficulty to understand this sentence "During 2012-2017 ... marine invertebrates,". Lines 26-30, should rewrite into a more focus fact, where the discovery of marine natural products from invertebrates has been increasing.

Response:

Thank you very much for your comments. We have revised the sentences presented in paragraph 1 as follows; “During 2012–2017, no less than 550–700 new compounds have been reported from marine invertebrates [4], in which half of these compounds were isolated from marine sponges [5]. Among those, 4% and 22% of the compounds were identified in 2017 and 2016, respectively [4,5]. Between 1998 to 2018, one hundred and fourteen secondary metabolites were isolated from the marine sponges of the genus Suberea [6]. Mean-while, a hundred and seventy compounds were isolated from soft corals of the genus Dendronephthya alone throughout 1999-2019 [7]. Soft corals belonging to the genus Xenia are rich in terpenoids, with 199 compounds isolated from 1977 to 2019 [8]. To date, approximately 30,000–40,000 marine natural products (MNPs) have been identified, with the majority of the compounds exhibiting cytotoxic and anticancer properties [4,5,9]”. We have also cited two additional references suggested by the reviewer in the comment above.

  1. 110-111 from Table 1, mollamides B-C might be categorised as peptide instead of cyclic hexapeptides. 

Response:

Thank you very much for your suggestion. We have revised 110-110 to be categorised as peptides instead of hexapeptides.

  1. Table 1, there are a lot of mistake when mentioned the compound name, for example "mollamide B and mollamide C" or "mollamides B and C". Another example is "chloroscabrolide A" not "chloroscabrolides A".

Response:

Thank you very much for your remarks. We have checked and corrected mistakes in Table 1 (now Tables 1-3 as per the suggestion of reviewer #2)

  1. Figure 7, compound 85 and 86 were not drew properly, please redraw it. The dashed line was reversed and OH bond bended. 

Response:

Thank you for your correction. We have corrected Figure 7 in the revised manuscript.

  1. Figure 7, compound 87 and 88 were not drew properly, the dashed line of Leu from the branch of cyclic peptide has reversed. Second, why only D-Me Val and D-Val were labelled? But amino acid residues were labeled in 89-94, Author have to consider consistency.

Response:

Thank you for your advice. We have revised Figure 7 and rechecked for consistency on the figures presented in the present manuscript.

  1. Figure 9, please enlarge your chemdraw template, as you can see the NH2 has squeezed in 99.

Response:

Thank you very much for pointing this out. Figure 9 have been revised so that NH2 can be adequately seen.

  1. Figure 13 (B) and (C) are difficult to see, because small and blurred.

Response:

Thank you very much for the comment. We have revised Figure 13 and increased the figure's resolution to make it easier to see.

Reviewer 2 Report

The review entitled “Chemical Diversity and Pharmacological Activity of Organic Compounds Isolated from Indonesian Marine Invertebrates” aims at reporting the natural products isolated from three Indonesian marine invertebrates, including sponges, tunicates, and soft corals,  covering the literature published between 2007 and 2019. A total of 124 compounds are presented and their biological activities are also discussed.

This manuscript falls in the scope of Molecules. It is well written and presents a significant scientific contribution. However, some minor revisions and corrections are needed before being accepted for publication.

  1. Biological activity and pharmacological activity are not quite the same. What is described in the paper are, in most of the examples, preliminary biological activities. Therefore, for accuracy reasons, this must be corrected throughout the paper, including the title, table, and the main text (the correct phrase is biological activity).
  2. In the title it is also better to change organic compounds for secondary metabolites; In this way, the title should be changed to Chemical Diversity and Biological Activity of secondary metabolites Isolated from Indonesian Marine Invertebrates
  3. Line 38 – Correct: EMA – European Medicines Agency
  4. Line 41 - Omega-3-acid-ethyl esters derived from fish oil
  5. Table 1 is very large. It should be divided into 3 different tables comprising natural products isolated from sponges (Table 1), Tunicates (Table 2), and soft coral (Table 3), in accordance with the text. In this way, column 4 should be deleted. Moreover, it is better that the number of the compounds appear after the name of the compound and not in a separated column, for example, Acanthomanzamine A (1).
  6. The names of compounds 1 – 6 should be corrected in Table 1.
  7. Acanthocyclamine A (8) has an inhibitory effect on amyloid B-42 production rather than anti-Alzheimer activity. Please correct in Table 1.
  8. Table 1 – cell lines should be described in the table and in the text. For example, what type of cells are A431 and L5178Y.
  9. Authors report the biological activity both is µg/mL and µM. Although µg/mL is frequently used to assess the activity of isolated compounds, this reviewer considered that it is not the most correct way. Therefore, µM concentrations should be also added in order to harmonize the provided information.
  10. Lines 177 – 185 – There´s no need to describe here the NMR and MS data of this compound. These lines should be deleted.
  11. Lines 197 – 199 – delete the numbers (1, 2, 3 and 4) that appear before the names of compounds (ex: … namely (1) 11-methoxy-3H-[1,6] naphthyridino [6,5,4-ef]quinoxalin-3- (46), (2) 2,11-dimethoxy-3H-[1,6] naphthyridino [6,5,4-def]-quinoxalin-3-one (47), (3) 5-benzoyldemethylaaptamine (48), and (4) 3-aminodemethyl(oxy)aaptamine (49)
  12. 11-methoxy-3H-[1,6] naphthyridino [6,5,4-ef]quinoxalin-3- ? Is this name correct?
  13. Line 258 – correct: 1,1-diphenyl-2-picrylhydrazyl radical (DPPH)
  14. Correct the legend of Figure 7. I believe that the two sentences belong to the main text.
  15. Line 315: “Compound 103 is the enantiomer of the previously isolated compound”. To what compound do you refer?
  16. Lines 335 – 338: “Furthermore, compound 111 showed anticancer activity with a unit zone differential value of 100 against L1210, human colon HCT-116, and human lung H125 cells and a value of 250 against murine colon 38”. What kind of values do you refer to? Differential values of 100?

Author Response

Reviewer #2:

General comments

The review entitled "Chemical Diversity and Pharmacological Activity of Organic Compounds Isolated from Indonesian Marine Invertebrates" aims at reporting the natural products isolated from three Indonesian marine invertebrates, including sponges, tunicates, and soft corals, covering the literature published between 2007 and 2019. A total of 124 compounds are presented and their biological activities are also discussed.

This manuscript falls in the scope of Molecules. It is well written and presents a significant scientific contribution. However, some minor revisions and corrections are needed before being accepted for publication.

Specific comments

  1. Biological activity and pharmacological activity are not quite the same. What is described in the paper are, in most of the examples, preliminary biological activities. Therefore, for accuracy reasons, this must be corrected throughout the paper, including the title, table, and the main text (the correct phrase is biological activity).
  2. In the title it is also better to change organic compounds for secondary metabolites; In this way, the title should be changed to Chemical Diversity and Biological Activity of secondary metabolites Isolated from Indonesian Marine Invertebrates

Response:

Thank you very much for the suggestion. We have changed the title to "Chemical Diversity and Biological Activity of Secondary Metabolites Isolated from Indonesian Marine Invertebrates" as suggested. The related phrases also have been corrected throughout the paper.

  1. Line 38 – Correct: EMA – European Medicines Agency

Response:

Thank you very much for the correction. The abbreviated form of the European Medicines Agency has been corrected from "EMEA" to "EMA".

  1. Line 41 - Omega-3-acid-ethyl esters derived from fish oil

Response:

Thank you very much for the pointer. We have corrected it to "Omega-3-acid-ethyl esters derived from fish oil".

  1. Table 1 is very large. It should be divided into 3 different tables comprising natural products isolated from sponges (Table 1), Tunicates (Table 2), and soft coral (Table 3), in accordance with the text. In this way, column 4 should be deleted. Moreover, it is better that the number of the compounds appear after the name of the compound and not in a separated column, for example, Acanthomanzamine A (1).

Response:

Thank you very much for your suggestions. We have divided Table 1 into three tables (Tables 1-3) and the columns have been reorganised as necessary.

  1. The names of compounds 1 – 6 should be corrected in Table 1.

Response:

Thank you very much for your remarks. The names of compounds 1 – 6 in Table 1 have been rechecked and corrected carefully.

  1. Acanthocyclamine A (8) has an inhibitory effect on amyloid B-42 production rather than anti-Alzheimer activity. Please correct in Table 1.

Response:

Thank you very much for the suggestion. We have revised the sentence as follows: “inhibitor of amyloid β-42 production” in Table 1.

  1. Table 1 – cell lines should be described in the table and in the text. For example, what type of cells are A431 and L5178Y.

Response:

Thank you very much for the comment. The information of the cell lines has been added both in the tables and in the text.

  1. Authors report the biological activity both is µg/mL and µM. Although µg/mL is frequently used to assess the activity of isolated compounds, this reviewer considered that it is not the most correct way. Therefore, µM concentrations should be also added in order to harmonise the provided information.

Response:

Thank you very much for the suggestion. We have revised and harmonised the unit of biological activities. However, activities of inseparable mixture and unit of differential values cannot be transformed into µM because there is not enough information available on the original paper.

  1. Lines 177 – 185 – There's no need to describe here the NMR and MS data of this compound. These lines should be deleted.

Response:

Thank you very much for your suggestions. We have deleted the NMR and MS data in the revised manuscript.

  1. Lines 197 – 199 – delete the numbers (1, 2, 3 and 4) that appear before the names of compounds (ex: … namely (1) 11-methoxy-3H-[1,6] naphthyridino [6,5,4-ef]quinoxalin-3- (46), (2) 2,11-dimethoxy-3H-[1,6] naphthyridino [6,5,4-def]-quinoxalin-3-one (47), (3) 5-benzoyldemethylaaptamine (48), and (4) 3-aminodemethyl(oxy)aaptamine (49)

Response:

Thank you very much for your remarks. We have deleted the numbers (1, 2, 3 and 4) that appear before the names of compounds in lines 197-199 in the revised manuscript (lines 191-193 in the revised manuscript).

  1. 11-methoxy-3H-[1,6] naphthyridino [6,5,4-ef]quinoxalin-3- ? Is this name correct?

Response:

Thank you very much for noticing this. The name of the compound has been corrected into “11-methoxy-3H-[1,6] naphthyridino [6,5,4-ef]quinoxalin-3-one”.

  1. Line 258 – correct: 1,1-diphenyl-2-picrylhydrazyl radical (DPPH)

Response:

Thank you very much for your remarks. The abbreviation of DPPH has been corrected.

  1. Correct the legend of Figure 7. I believe that the two sentences belong to the main text.

Response:
Thank you very much for your suggestion. We have revised the two sentences in the legend of Figure 7.

  1. Line 315: "Compound 103 is the enantiomer of the previously isolated compound". To what compound do you refer?

Response:
Thank you very much for your remarks. The correct phrase would be "Compound 107 (previously compound 103) is the enantiomer of the previously isolated compound, (+)-leptoclinidamine B." It has been corrected in the revised manuscript.

  1. Lines 335 – 338: "Furthermore, compound 111 showed anticancer activity with a unit zone differential value of 100 against L1210, human colon HCT-116, and human lung H125 cells and a value of 250 against murine colon 38". What kind of values do you refer to? Differential values of 100?

Response:

Thank you very much for your remarks. Yes, the values referred to in the article are the differential values and these have been corrected. However, we cannot transform these differential values into more familiar units as there is not enough information available on the original paper.

Round 2

Reviewer 1 Report

Dear Authors,

Thank you for addressing the comments positively.

There is a minor point, Table 1, "Ethyl 3,4-dibromo-1H-pyrrole-2-carboxylate (2) (23)" should be Ethyl 3,4-dibromo-1H-pyrrole-2-carboxylate (23).

Table 1, compounds 14-19, example (D-Pro-L-Phe) (13) should be (D-Pro-L-Phe) (13), the configuration of L/D amino acid should have smaller front size. 

Table 1, compounds 24-25, compounds 14-19, example 2-N-methyl stevensine (24) should be 12-N-methyl stevensine (24), where N should be italic.

Table 1, 11-epi-O,O-dimethyllingshuiolide A (75) should be 11-epi-O,O-dimethyllingshuiolide A (75), where epi should be italic.

Figure 3, R = OSO3H (36) should be R = OSO3H (36).

Figure 4, please check the structure 44 again.

Figure 11, structures should be standardize, it was obvious that structure's size of 118 is smaller than 119, although both shared a lot similar in their structure.

Author Response

Dear editors and reviewers,

Below are our responses to the comments of reviewer #1 for the review report round 2. We have copied and pasted their comments. The reviewer's comments appear in italics and follow in the same order as written by the referee. Our responses appear in regular font.

Reviewer #1: 

Thank you for addressing the comments positively.

There is a minor point, Table 1, "Ethyl 3,4-dibromo-1H-pyrrole-2-carboxylate (2) (23)" should be Ethyl 3,4-dibromo-1H-pyrrole-2-carboxylate (23).

Response:

Thank you very much for your correction. "Ethyl 3,4-dibromo-1H-pyrrole-2-carboxylate (2) (23)" has been corrected to “Ethyl 3,4-dibromo-1H-pyrrole-2-carboxylate (23)”.

Table 1, compounds 14-19, example (D-Pro-L-Phe) (13) should be (D-Pro-L-Phe) (13), the configuration of L/D amino acid should have smaller front size.

Response:

Thank you very much for your advice. We have revised the L/D amino acid configuration in Table 1 with smaller font size and have made these corrections:

  • "Cyclo (D-Pro-L-Phe) (13)" has been corrected to "Cyclo (d-Pro-l-Phe) (13)"
  • "Cyclo (L-Pro-Gly) (14)" has been corrected to "Cyclo (l-Pro-Gly) (14)"
  • "Cyclo (L-Pro-L-Ala) (15)" has been corrected to "Cyclo (l-Pro-l-Ala) (15)"
  • "Cyclo (D-Pro-L-Val) (16)" has been corrected to "Cyclo (d-Pro-l-Val) (16)"
  • "Cyclo (L-Pro-Ser) (17)" has been corrected to "Cyclo (l-Pro-Ser) (17)"
  • "Cyclo (D-Pro- L-Ile) (18)" has been corrected to "Cyclo (d-Pro- l-Ile) (18)"
  • "Cyclo (L-Pro- L -Tyr) (19)" has been corrected to "Cyclo (l-Pro- l -Tyr) (19)"

Table 1, compounds 24-25, compounds 14-19, example 2-N-methyl stevensine (24) should be 12-N-methyl stevensine (24), where N should be italic.

Response:

Thank you very much for your comment. We have corrected the N in compounds 24-25 from Table 1 to be italics and revised "12-N-methyl stevensine (24)" to "12-N-methyl stevensine (24)" and "12-N-methyl-2-debromostevensine (25)" to "12-N-methyl-2-debromostevensine (25)."

Table 1, 11-epi-O,O-dimethyllingshuiolide A (75) should be 11-epi-O,O-dimethyllingshuiolide A (75), where epi should be italic.

Response:

Thank you very much for your correction. "11-epi-O,O-dimethyllingshuiolide A (75)" has been corrected to "11-epi-O,O-dimethyllingshuiolide A (75)".

Figure 3, R = OSO3H (36) should be R = OSO3H (36).

Response:

Thank you very much for noticing this. R = OSO3H (36) has been corrected to R = OSO3H (36) on Figure 3.

Figure 4, please check the structure 44 again.

Response:

Thank you very much for your remark. We have checked the structure 44 according to the reference (Aoki et al., Tetrahedron Letters, 48 (26), 2007, 4485-4488, 10.1016/j.tetlet.2007.05.003) and it is found that the structure is correct.

Figure 11, structures should be standardize, it was obvious that structure's size of 118 is smaller than 119, although both shared a lot similar in their structure.

Response:
Thank you very much for your suggestion. We have revised the structures on Figure 11 as suggested
